# Assessment of Saudi Society’s Knowledge Regarding Hypothyroidism and Its Neuropsychiatric Clinical Manifestations

**DOI:** 10.3390/healthcare11020277

**Published:** 2023-01-16

**Authors:** Hayat Saleh Alzahrani, Rand Abdalla Alshabnan, Fatmah Mamdooh Mokhtar, Aljoharah Ibrahim Aleisa, Nora Abdulrahman AlHedaithi, Ghadah Khalid Alotaibi, Tif Meshref Alamri, Wejdan Dia Aluthaim, Jana Mahmoud Alyousef, Reem Saeed AlSarhan, Maha Mohammed AlHussein, Bader A. Almehmadi, Mansour Alzahrani, Fahad Mohammad Alfhaid

**Affiliations:** 1Department of Clinical Science, College of Medicine, Princess Nourah Bint Abdulrahman University, Riyadh 11671, Saudi Arabia; 2College of Medicine, Princess Nourah Bint Abdulrahman University, Riyadh 11671, Saudi Arabia; 3Department of Internal Medicine, College of Medicine, Majmaah University, Al-Majmaah 11952, Saudi Arabia; 4Department of Family and Community Medicine, College of Medicine, Majmaah University, Al-Majmaah 11952, Saudi Arabia

**Keywords:** assessment, knowledge, Saudi population, hypothyroidism, Saudi Arabia, depression, anxiety, cross-sectional study

## Abstract

Background: This study was conducted to assess the level of knowledge and awareness of hypothyroidism and its neuropsychiatric clinical manifestations among the Saudi population. Methods: This was a cross-sectional study employing a convenient sampling technique, conducted between February and May 2022. A questionnaire was distributed online to all participants in all five regions. Results: In this survey, a total of 2016 Saudi citizens participated. When asked about depression, more than half of the participants (59.6%) correctly identified depression as one of the neuropsychiatric clinical symptoms of hypothyroidism. Nearly half of the participants (47.5%) were unaware that anxiety was not a neuropsychiatric manifestation of hypothyroidism. With a percentage of 91.0%, the majority of participants exhibited poor knowledge. The regression analysis showed that males have significantly reduced knowledge about hypothyroidism than females (coefficient −3.686, *p*-value < 0.0001). Similarly, those who have “enough income and can save” were more knowledgeable than others (coefficient 0.731, *p*-value < 0.02). Regarding the source of information, journals provide three times more information (*p*-value 0.0001), and healthcare practitioners provides four times more information as compared to family and friends (*p*-value 0.0001). Conclusion: Due to a lack of knowledge about hypothyroidism and its complications, symptoms, risk factors, and treatment, the most viable solution to these misconceptions would be to implement a variety of educational programs to increase public awareness of this issue.

## 1. Introduction

The thyroid gland plays a critical role in the body’s metabolism, growth, and development [1]. Any increase or decrease in the production of these hormones will result in observable changes [2]. The gland’s normal function is especially important for developing neurocognition and growth during adolescence, childhood, and adulthood, as well as for maintaining physiological functions.

Thyroid disorders are conditions associated with excessive or insufficient thyroid hormone secretion. Hypothyroidism is a disorder characterized by insufficient secretion; it is of two types: primary, which is related to the gland itself, and secondary, which is caused by other factors [3]. Clinical manifestations of hypothyroidism can involve major body systems such as the central nervous system, cardiovascular system, gastrointestinal system, reproductive system, and endocrine system, and include weight gain, fatigue, cold intolerance, slow heart rate, shortness of breath, constipation, and irregular menstrual cycles in females [4]. Since hypothyroidism affects multiple systems, these symptoms can be confused with those of other disorders, which explains why the condition is frequently misdiagnosed. Hypothyroidism is characterized by decreased circulating levels of one or both thyroid hormones or insufficient stimulation of thyroid-stimulating hormone (TSH), making thyroid function tests the ideal diagnostic tool [5].

Few publications discussed the association between hypothyroidism and neuropsychiatric effects. Nonetheless, according to a 2014 study, “clinical investigations and functional imaging studies confirm that overt hypothyroidism is associated with affective and cognitive deficits” [6]. According to another study, hypothyroidism affects cognitive functioning and mood. Severe hypothyroidism can resemble melancholic depression and dementia [7]. The purported relationship between hypothyroidism and neuropsychiatric clinical manifestations demonstrates the importance of assessing Saudi citizens’ knowledge of the relevance of hypothyroidism to neuropsychiatric symptoms, as these symptoms can interfere with patients’ daily lives and productivity. Insufficient evidence regarding hypothyroidism in the Saudi population necessitates further explanation.

Consequently, the purpose of this study was to assess the Saudi population’s level of knowledge regarding hypothyroidism and its manifestations, complications, and factors influencing that level of knowledge.

## 2. Materials and Methods

This cross-sectional, online questionnaire-based study assessed Saudi citizens’ knowledge of hypothyroidism. The survey’s online questionnaire was distributed to Saudi citizens and residents in five major regions of Saudi Arabia (Central, Eastern, Western, Northern, and Southern regions). A total of 2016 participants participated in the study. The data were collected from February 2022 to May 2022.

The inclusion criteria for the selection of subjects for this study included Saudi citizens who were ≥ 18 years old. No gender or geographic restrictions were enforced. The exclusion criteria included non-Saudis, Saudi citizens under 18, and those who could not consent.

The questionnaire was developed after the literature review and adapted from a previously published study [2,7,8]. The Cronbach’s alpha value mentioned in one of the studies was 0.7, which is acceptable and showed good homogeneity. Face validation of the instrument was determined by the group of expert researchers (faculty members) after they reviewed the questionnaire content. A back-to-back Arabic translation was performed until the contents of the questionnaire became clear and covered the study’s objectives. The finalized questionnaire version was uploaded as a Google Form, and an open link was generated.

The survey utilized a convenience sampling technique. In order to ensure that the survey was distributed to various regions in Saudi Arabia, the study investigators initially shared an unrestricted link to the survey via social networks (WhatsApp, Facebook, Twitter, etc.) with their primary contacts. The primary contacts were then asked to forward the survey to their connections, and so on.

The first part of the questionnaire includes information about sociodemographic data. The second part contains general knowledge questions regarding hypothyroidism, knowledge of hypothyroidism symptomology, knowledge of risk factors, the difference in prevalence between the sexes, complications and neuropsychiatric manifestations, and treatment. Answers such as “yes”, “no”, or “I don’t know” were used to assess the responses to each knowledge question.

Regarding the sample size, previous research indicated that the average Saudi population’s knowledge of thyroid dysfunction was approximately 43.91 percent, with a margin of error of 5 percent and a study power of 95 percent. In this study, the sample size was estimated to be 1056 using the G-power application, but in the end, a total of 2016 Saudi males and females aged ≥ 18 were recruited.

Using SPSS, the level of knowledge was determined. In addition, a scoring system was implemented that required participants to score at least 60% (24 questions) correctly to be considered knowledgeable. Frequency and percentages were reported. Univariate variate analysis was performed for an initial identification of associated variables. Ordinary Least Squares (OLS) regression was performed to recognize the factor associated with increased knowledge. Significant variables (*p*-value ≤ 0.05) were then computed for the regression model for exploring the adjusted effect.

## 3. Results

In this study, a total of 2016 individuals from the five regions of Saudi Arabia participated in the survey. The participants’ mean age was 38.86 ± 13.3 years, as shown in Table 1. The proportion of female participants was significantly higher (77.9%) than that of male participants (22.1%). The majority of responses (68.7%) came from the central region, while the lowest response rate (1.5%) was recorded in the northern region. In terms of marital status, more than half of the participants (62.2%) were married. Regarding educational level, the highest response rate (70.9%) came from university graduates. With a percentage of (56.6%), nearly half of the participants’ income was adequate. Most participants (93.4%) were employed outside the healthcare industry.

As shown in Table 2, when assessing the participants’ general knowledge of hypothyroidism, 75.4% of participants correctly identified low thyroid hormone levels as the cause of hypothyroidism. Regarding the communicability of hypothyroidism, 94.9 percent of respondents correctly identified that it is not a communicable disease. A total of 59.7% were aware that females are more susceptible to hypothyroidism, as indicated by the correct response. When asked if a positive family history increases the risk of developing hypothyroidism, more than half of the participants (56.3%) responded correctly. Regarding decreased iodine intake as a cause of hypothyroidism, 49 percent of respondents answered “I don’t know,” while 46.7% answered correctly. Smoking and the presence of autoimmune diseases are risk factors for hypothyroidism, but 58.6% and 62.9%, respectively, were unaware of this fact when their knowledge of risk factors was evaluated (Figure 1). A total of 54.3 percent of participants correctly identified cold intolerance as a symptom of hypothyroidism. In contrast, more than half of the participants (53.9 percent) answered “I don’t know” when asked whether constipation is a symptom of hypothyroidism. Most respondents (81.1%) correctly identified weight gain as a symptom of hypothyroidism. Regarding the complications of hypothyroidism, 61.1% of respondents answered “I don’t know” when asked if thyroid cancer is a complication of hypothyroidism, whereas only 21.5% answered correctly. Evaluation of whether hypothyroidism causes menstrual irregularities revealed that sixty-three percent of respondents were aware of this fact. As for the presence of an elevated level of cholesterol in the blood due to hypothyroidism, it was discovered that 58.5% of respondents responded with “I don’t know”, and only 30.5% correctly identified this statement. Moving on to the question of whether hypothyroidism is linked to nerve damage, it was discovered that 64.4% of respondents answered “I don’t know”, and only 18% answered “yes”. Regarding the neuropsychiatric symptoms of hypothyroidism, overall, participants mostly do not know its manifestation; however, almost 60% of the participants correctly identified that depression is one of the symptoms of hypothyroidism (Figure 2). For hypothyroidism treatment options, the majority (80.2%) of respondents knew that thyroid-stimulating oral medication was a treatment option.

Table 3 displays the mean knowledge score according to the independent variables. It was determined that (91%) of the participants had inadequate knowledge. Further analysis of the total knowledge score revealed, with a *p*-value of 0.037, that the 18–26 age group had a higher knowledge score (16.227.15) than other age groups. As for the knowledge score between genders, females have a higher score than males (16.446.22), with a *p*-value of 0.001, indicating a statistically significant difference. With a *p*-value of 0.012, a disparity in income knowledge was observed, with sufficient income and the ability to save receiving the highest score (16.217.08), and inadequate income and debt receiving the lowest score (13.907.57). Surprisingly, the *p*-value of 0.49 indicated no significant difference between healthcare workers (20.247.14) and nonhealthcare workers (15.276.69).

Table 4 shows Ordinary Least Squares (OLS) regression analysis. The regression analysis showed that increases in age were inversely related to knowledge, meaning that as age increases, the knowledge about hypothyroidism decreases, although this relationship is not significant. Likewise, males have significantly reduced knowledge about hypothyroidism than females (coefficient −3.686, *p*-value < 0.0001). Similarly, those who have “enough income and can save” were more knowledgeable than others (coefficient 0.731, *p*-value < 0.02). Regarding the source of information, journals provide three times more information as compared to family and friends (*p*-value 0.0001), and healthcare practitioners provide four times more information as compared to family and friends (*p*-value 0.0001).

## 4. Discussion

It is essential to raise awareness of hypothyroidism and illuminate the community’s level of knowledge to influence people’s thoughts and actions.

A study conducted in Croatia shows that the prevalence of hypothyroidism disorder in their population is 10.5% [9]. According to another study conducted in Jordan, the prevalence of hypothyroidism is 17.2% in females and 9.1% in males [10]. Given these alarming prevalence percentages, it is crucial to influence the acceptance of the proposed preventative measures to increase awareness of hypothyroidism.

In this study, 2016 participants from the five regions of Saudi Arabia, including 77.9% female and 22.1% male participants, provided data. The calculated scores for knowledge were 9 percent good knowledge and 91 percent poor knowledge.

A 2021 study had a poor knowledge score of 28.2% [7]. Their level of knowledge was greater than ours, which could be attributed to the fact that nearly half of their study participants (n = 787, 50.4%) had previously been diagnosed with a thyroid disorder or had a family member with the condition. This variable was significantly associated with their study’s overall knowledge score (*p =* 0.001). In a 2020 study, the overall knowledge score was 44.7% [11]. Compared to our research, their results demonstrated a higher level of knowledge, which may be because our study was more focused on hypothyroidism than general thyroid disorders. In addition, a substantial proportion of their subjects had previously been diagnosed with hypothyroidism and undergone thyroid gland examinations.

There were a number of factors that influenced the knowledge level of our participants, with the first being that the majority of respondents were females (77.9%), which could be explained by the higher prevalence of hypothyroidism disease among females, potentially making them more interested in the topic. In a study of 3872 participants conducted at King Fahad Armed Forces Hospital in Jeddah, Saudi Arabia, 1125 individuals were identified as having hypothyroidism. Of these 1125 individuals, 964 (85.7%) were female, while only 161 (14.3%) were male [12].

The other factor that should be evaluated is the participants’ high socioeconomic status. This factor was anticipated to have a positive effect on the knowledge score for two possible reasons. First, with a higher socioeconomic status, family members have access to better education and to healthcare facilities with higher standards, where the staff is more attentive to raising patient awareness. Another factor to be considered is the origin of the data.

More than half of the participants (58.4%) cited family and friends as their primary source of knowledge, which may have contributed to the decline in knowledge. Participants aged between 18 and 26 years had the highest knowledge score, which could be attributed to the fact that they had greater access to online information. The fact that the quality of education is improving over time so that the younger generation has greater access to information may also play a role, as the current emphasis on health-related information causes individuals to become more aware of chronic conditions.

Multiple notable misunderstandings may have contributed to the inadequate knowledge of our respondents. First, regarding the symptoms of hypothyroidism, only 12.2% identified that anxiety is not a symptom of hypothyroidism, whereas 59.9% were able to do so in a study conducted in 2021 [7]. Our participants may have exhibited this low percentage because neuropsychiatric manifestations of hypothyroidism are not distinguished from neuropsychiatric manifestations of hyperthyroidism.

In our study, 12.9% agreed that smoking is a risk factor for hypothyroidism. In contrast, in another study conducted in the eastern province of Saudi Arabia, a higher percentage of 45.5% agreed with this statement [11]. This knowledge gap may be because our questionnaire distribution was limited to the central region, whereas the other study was limited to the eastern region. The use of oral contraceptive pills (OCPs) is another risk factor that must be evaluated; in our study, 22.6% of participants reported that OCP use is a risk factor for hypothyroidism.

However, according to a study published in the British Medical Journal, taking OCPs for one month to one year increases the risk of developing hypothyroidism by 17.4% [12]. One possible explanation for the low awareness of this risk factor is that the topic of OCPs may be viewed as taboo, resulting in less overall discussion about the side effects of OCPs; consequently, individuals may be unaware of the side effects.

Thirdly, in terms of the complications of hypothyroidism, 30.5% of our participants were aware that elevated blood cholesterol levels are a complication of hypothyroidism, which is lower than the 62.4% of Saudi Arabian participants who were aware of this fact [6]. The difference between the two studies may be attributed to our study assessing knowledge regarding hypothyroidism specifically, whereas the previously cited study assessed knowledge regarding all thyroid disorders.

Regarding the treatment of hypothyroidism, nearly half of the participants (46.4%) believed that the thyroid gland could be surgically removed to treat the condition.

According to the Columbia Thyroid Center, however, surgical thyroid gland removal will result in hypothyroidism [13,14]. This error in theunderstanding of participants may have confused hyperthyroidism and hypothyroidism management, respectively. According to the National Library of Medicine, thyroidectomy is used for the long-term management of hyperthyroidism, which is the most common complication of this surgery [15].

The data were collected via an online questionnaire form, which was a limitation because the survey distribution was limited to individuals with access to electronic devices. Another limitation of this study is that it was not uniformly distributed across all Saudi regions. This survey may have also been affected by a sampling bias in which only individuals interested in the topic participated. The data were collected based on a self-reporting method to assess the knowledge; this might pose some issues related to the accuracy of information. The study’s main strength is that all Saudi citizens were eligible irrespective of their region or disease status. Secondly, we included a large number data to overcome the issue of missing data. Duplicate forms were removed. We ensured that the generated questionnaire link could be used one time for one IP address. In this way, we tried to overcome the issue of participation duplication.

## 5. Conclusions

In conclusion, the findings of this study indicated that Saudi participants possessed a limited understanding of hypothyroidism and a misunderstanding of its neuropsychiatric symptoms. Our study shows that females with enough income and the ability to save income have more knowledge about hypothyroidism as compared to others. Likewise, healthcare practitioners and journals are the main source of information. These data shed light on the level of knowledge to facilitate the implementation of effective national-level campaigns and programs and will increase the level of knowledge and awareness in the population. This will also positively impact the population’s commitment to the treatment of hypothyroid patients and the prevention of the disease in healthy individuals, thereby benefiting both the people and the government.

## Figures and Tables

**Figure 1 healthcare-11-00277-f001:**
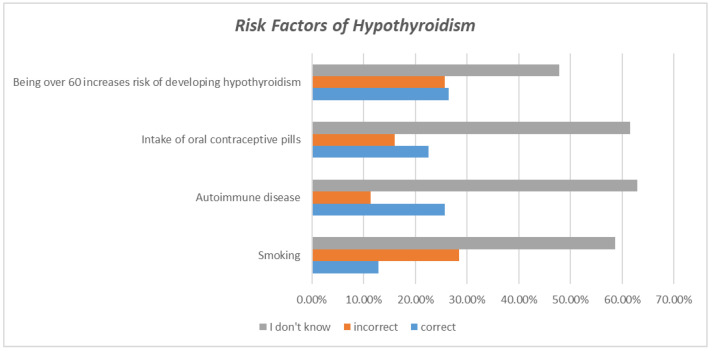
Risk factors of hypothyroidism.

**Figure 2 healthcare-11-00277-f002:**
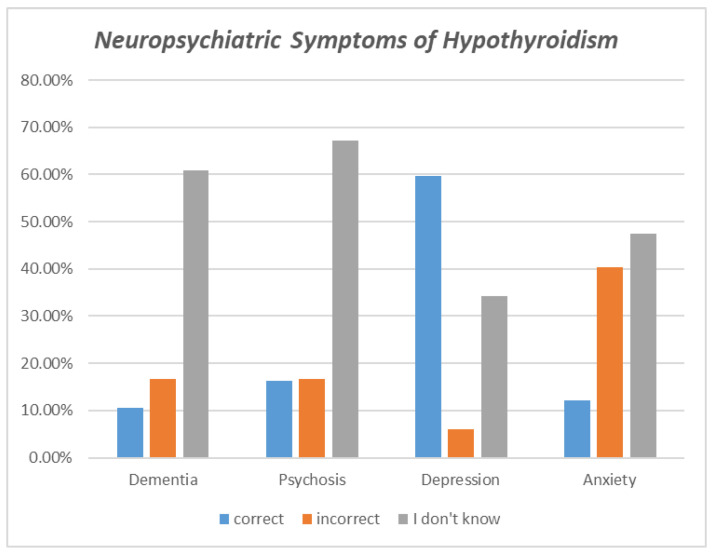
Neuropsychiatric symptoms of hypothyroidism.

**Table 1 healthcare-11-00277-t001:** Demographic characteristics of the study sample, n = 2016.

Variables	n (%)
Age (years)	38.86 ± 13.84
Gender	
Females	1571 (77.9%)
Males	445 (22.1%)
Region	
Central region	1384 (68.7%)
Eastern region	145 (7.2%)
Western region	253 (12.5%)
Northern region	31 (1.5%)
Southern region	203 (10.1%)
Marital status	
Married	1254 (62.2%)
Unmarried	762 (37.8%)
Educational level	
School education	378 (18.8%)
University	1430 (70.9%)
Postgraduate	208 (10.3%)
Income	
Enough and can save	666 (33%)
Enough	1141 (56.6%)
Not enough	144 (7.1%)
Not enough/in debt	65 (3.2%)
Resources of information	
Health care practitioners	218 (10.8%)
Family/Friends	1179 (58.4%)
Social media platforms	217 (10.8%)
Scientific and Medical Journals/Books	404 (20%)
Health care worker?	
Yes	133 (6.6%)
No	1883 (93.4%)

**Table 2 healthcare-11-00277-t002:** Items for knowledge assessment.

Items for Knowledge Assessment	Correct n (%)	Incorrect n (%)	I Do Not Know n (%)
General Knowledge			
Hypothyroidism is caused by reduced levels of thyroid hormone levels	1521 (75.4%)	74 (3.7%)	421 (20.9%)
The ability to live without the thyroid gland	966 (47.9%)	508 (25.2%)	542 (26.9%)
Hypothyroidism is a communicable disease	1913 (94.9%)	8 (0.4%)	95 (4.7%)
Males are more likely to develop hypothyroidism	1156 (57.3%)	58 (2.9%)	802 (39.8%)
Females are more likely to develop hypothyroidism	1204 (59.7%)	111 (5.5%)	701 (34.8%)
Those with a positive family history are more likely to develop hypothyroidism	1136 (56.3%)	318 (15.8%)	562 (27.9%)
Decreased iodine intake leads to hypothyroidism	942 (46.7%)	87 (4.3%)	987 (49%)
Children can develop hypothyroidism	943 (46.8%)	217 (10.8%)	856 (42.5%)
Risk Factors of Hypothyroidism			
Smoking	260 (12.9%)	575 (28.5%)	1181 (58.6%)
Autoimmune disease	519 (25.7%)	229 (11.4%)	1268 (62.9%)
Intake of oral contraceptive pills	455 (22.6%)	322 (16.0%)	1239 (61.5%)
Being over 60 increases the risk of developing hypothyroidism	534 (26.5%)	518 (25.7%)	964 (47.8%)
Neuropsychiatric Symptoms of Hypothyroidism			
Dementia	212 (10.5%)	334 (16.6%)	1225 (60.8%)
Psychosis	329 (16.3%)	334 (16.6%)	1353 (67.1%)
Depression	1202 (59.6%)	122 (6.1%)	692 (34.3%)
Anxiety	245 (12.2%)	814 (40.4%)	957 (47.5%)
Symptoms of Hypothyroidism			
Cold intolerance	1095 (54.3%)	194 (9.6%)	727 (36.1%)
Constipation	607 (30.1%)	322 (13.0%)	1087 (53.9%)
Increased sweating	437 (21.7%)	688 (34.1%)	891 (44.2%)
Increased need to urinate	569 (28.2%)	286 (14.2%)	1161 (57.6%)
Palpitations	383 (19.0%)	693 (34.4%)	940 (46.6%)
Swelling of the neck	1224 (60.7%)	240 (11.9%)	552 (27.4%)
Weight gain	1634(81.1%)	100 (5%)	282 (14%)
Hand tremors	399 (19.8%)	534 (26.5%)	1083 (53.7%)
Complications of Hypothyroidism			
Thyroid cancer	433 (21.5%)	352 (17.5%)	1231 (61.1%)
Irregularities of the menstrual cycle	1215 (60.3%)	352 (17.5%)	716 (35.5%)
Increased levels of cholesterol in the blood	614 (30.5%)	222 (11%)	1180 (58.5%)
Osteoporosis	250 (12.4%)	688 (34.1%)	1078 (53.5%)
Diabetes	514 (25.5%)	365 (18.1%)	1137 (56.4%)
Nerve damage	363 (18%)	355 (17.6%)	1298 (64.4%)
Infertility	738 (36.6%)	286 (14.2%)	992 (49.2%)
Sight issues	443 (22%)	374 (18.6%)	1199 (59.5%)
Excessive body hair in females	415 (20.6%)	570 (28.3%)	1031 (51.1%)
Hypothyroidism Treatment			
Biopsy must be performed to diagnose hypothyroidism	976 (48.4%)	438 (21.7%)	602 (29.9%)
Thyroid-stimulating oral medication	1616 (80.2%)	45 (2.2%)	355(17.6%)
Surgical removal of the thyroid gland	935 (46.4%)	432 (21.4%)	649 (32.2%)

**Table 3 healthcare-11-00277-t003:** Average knowledge scores, mean, and standard deviation (SD).

Variables	Mean ± SD	*p*-Value
Age (years)		
18–26 (n = 567)	16.22 ± 7.15	0.037 *
27–40 (n = 491)	15.32 ± 6.95	
41 and above (n = 958)	15.37 ± 6.83	
Gender		
Females (n = 1571)	16.44 ± 6.22	<0.001 *
Males (n = 445)	12.61 ± 7.96	
Region		
Central region (n = 1384)	15.76 ± 6.86	0.177
Eastern region (n = 145)	14.74 ± 6.37	
Western region (n = 253)	14.91 ± 7.24	
Northern region (n = 31)	15.42 ± 7.01	
Southern region (n = 203)	15.98 ± 6.36	
Marital status		
Married (n = 1254)	15.42 ± 6.65	0.110
Unmarried (n = 762)	15.90 ± 7.12	
Educational level		
No education (n = 20)	14.95 ± 6.53	0.562
School education (n = 358)	15.34 ± 6.86	
University (n = 1430)	15.59 ± 6.77	
Higher education (n = 208)	16.15 ± 7.24	
Income		
Enough and can save (n = 666)	16.21 ± 7.08	0.012 *
Enough (n = 1141)	15.33 ± 6.61	
Not enough (n = 144)	15.65 ± 6.91	
Not enough/ in debt (n = 65)	13.90 ± 7.57	
Source of information		
Healthcare practitioners (n = 218)	18.8 ± 6.5	<0.001 *
Family and Friends (n = 1179)	14.5 ± 6.2	
Social media platforms (n = 217)	14.3 ± 7.3	
Scientific and Medical Journals/Books (n = 404)	17.8 ± 7.5	
Health care worker		
Yes (n = 133)	20.24 ± 7.14	0.494
No (n = 1883)	15.27 ± 6.69	

* Significant difference.

**Table 4 healthcare-11-00277-t004:** Regression analysis.

Variables	Coefficients	Standard Error	t Stat	*p*-Value	Confidence Interval	
					Lower 95%	Upper 95%
Age						
18–26	Reference					
27–40	−0.054	0.403	−0.134	0.893	−0.844	0.736
46 and up	−0.066	0.347	−0.192	0.848	−0.748	0.615
Gender						
Female	Reference					
male	−3.686	0.349	−10.564	0.0001 *	−4.370	−3.002
Income						
Enough	Reference					
Enough and can save	0.731	0.315	2.32	0.02 *	0.113	1.349
Not enough	0.229	0.571	0.402	0.688	−0.891	1.35
In debt	−1.143	0.821	−1.393	0.164	−2.754	0.467
Source of information						
Family and Friends	Reference					
Healthcare practitioners	4.092	0.477	8.585	0.0001 *	3.158	5.027
Social media platforms	−0.2481	0.478	−0.519	0.604	−1.185	0.689
Scientific and Medical Journals/Books	3.101	0.375	8.271	0.0001 *	2.366	3.836

* Significant difference.

## Data Availability

Not applicable.

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
