# Peer review of "Assessment of Saudi Society’s Knowledge Regarding Hypothyroidism and Its Neuropsychiatric Clinical Manifestations"

_healthcare, 2023, doi:10.3390/healthcare11020277_

Round 1
Reviewer 1 Report
Thanks for the invitation. This study focuses on one of the ongoing hot topics, but some modifications need to be made. Regression can be added.
Other comments are:
The study design: is insufficient, increase length with more specific details.
For analyzing participant's data from online database, several factors are always difficult to overcome, including data missing, poor qualification, participation duplication and so on.
please add the strength and limitations of this study
In the methods, mention it is an online questionnaire survey. Mention the validity and reliability of the questionnaire and how it was measured.
Which social media platform was used to distribute the questionnaire?
From which city or district the participants were selected?
Revise the conclusion with consistency to the revised manuscript
Author Response
Dear Reviewer,
Thank you for the suggestions, we have addressed all the comments in the manuscript with track changes and yellow highlights.
Regards
Comments and their responses:
Thanks for the invitation. This study focuses on one of the ongoing hot topics, but some modifications need to be made. Regression can be added.
Regression analysis added in the result section.
Other comments are:
The study design: is insufficient, increase length with more specific details.
Thank you for the suggestion. Changes have been Done
For analyzing participant's data from online database, several factors are always difficult to overcome, including data missing, poor qualification, participation duplication and so on.
please add the strength and limitations of this study
Changes have been made.
In the methods, mention it is an online questionnaire survey. Mention the validity and reliability of the questionnaire and how it was measured.
Changes have been Done as per suggestion
Which social media platform was used to distribute the questionnaire?
It has been added in methodology section.
From which city or district the participants were selected?
It has been added in methodology section
Revise the conclusion with consistency to the revised manuscript
Revised.
Reviewer 2 Report
Dear Author
Your research is an interesting one. Some parts should be improved before publication
1- English edit is required.
2- The ethical committee's name and registration code are needed.
3- Results are presented by the table and I think one or two figures are required to improve the manuscript "A picture is worth a thousand words"
Author Response
Dear Reviewer,
Thank you for the suggestions, we have addressed all the comments in the manuscript with track changes and yellow highlights.
Regards
Comments and Responses:
1- English edit is required.
Thank you for the suggestion, the English language editing have been made throughout the manuscript.
2- The ethical committee's name and registration code are needed.
Added
3- Results are presented by the table and I think one or two figures are required to improve the manuscript "A picture is worth a thousand words".
Added